# Histological changes of female reproductive organs subjected to different jumping exercise intensities and honey supplementation in rats

Maryam Mosavat[1], Mahaneem Mohamed[2], Foong Kiew Ooi[1,3], Mitra Mirsanjari[4,5], Anani Aila Mat Zin[6] and Aminah Che Romli[2]

[1] Sport Science Unit, School of Medical Sciences, Universiti Sains Malaysia, Kubang Kerian, Malaysia
[2] Department of Physiology, School of Medical Sciences, Universiti Sains Malaysia, Kubang Kerian, Malaysia
[3] Exercise and Sports Science Programme, School of Health Sciences, Universiti Sains Malaysia, Kubang Kerian, Malaysia
[4] Nutrition Programme, School of Health Sciences, Universiti Sains Malaysia, Kubang Kerian, Malaysia
[5] Mazandaran University of Medical Sciences, Emam Khomeini Hospital, Fereidonkenar, Mazandaran, Iran
[6] Department of Pathology, School of Medical Sciences, Universiti Sains Malaysia, Kubang Kerian, Malaysia

## ABSTRACT

**Background:** We assessed histopathological changes of ovaries and uterus in female rats subjected to different jumping exercise intensities combined with honey supplementation at one g/kg body weight/day.
**Methods:** A total of 72 rats were divided into six groups, 12 rats in each: control (C), 20 and 80 jumps (20E, 80E), honey (H), and 20 and 80 jump with honey (20EH, 80EH).
**Results:** The endometrium was significantly thicker in the rats in H, 20EH and 80EH groups compared to C, 20E, and 80E. The myometrium thickness was significantly lower in 80E and significantly higher in 80EH compared to C, respectively. There was significantly higher myometrium thickness in 20EH and 80EH compared to 20E and 80E and H. The number of glands of the uterus in 20E and 80E was significantly lower than C. However, there was a significantly higher number of glands in H, 20EH, and 80EH compared to 20E and 80E. The numbers of uterus vessels were significantly lower in 80E compared to 20E. However, the numbers of vessels were significantly higher in H, 20EH, and 80EH compared to 80E. The number of ovarian haemorregia was significantly lower in 20E, 80E, H, 20EH, and 80EH compared to C. The number of corpora lutea was significantly lower in 80EH, H, 80E, and 20E compared to C. However, the number of corpora lutea was significantly higher in 20EH compared to J20 and H.
**Conclusion:** This study suggested that jumping exercises in particularly high-intensity exercise may induce histopathological changes in uterus and ovary in rats, and honey supplementation may ameliorate these effects.

Corresponding author
Foong Kiew Ooi, fkooi@usm.my

**How to cite this article** Mosavat M, Mohamed M, Ooi FK, Mirsanjari M, Mat Zin AA, Che Romli A. 2019. Histological changes of female reproductive organs subjected to different jumping exercise intensities and honey supplementation in rats. **PeerJ 7:e7646**
**DOI 10.7717/peerj.7646**

## INTRODUCTION

Apart from the important health benefits earned by exercise and physical activity, high-intensity physical activity may accompany with female reproductive disorders. Low energy availability and stress induced by intense exercise may cause hypothalamic dysfunction such as disturbance of gonadotropin realizing hormone pulsatility and hypoestrogenism which in turn may result in menstrual disorder, infertility and osteoporosis (*Warren & Perlroth, 2001*; *Mountjoy et al., 2014*; *De Souza & Williams, 2004*). On the other side, high intensity and prolonged physical activity increase production of reactive oxygen species (ROS) by metabolic and physiological processes and cause cellular damage such as lipid damage, depletion of adenosine triphosphate and inhibition of protein synthesis (*Powers, Radak & Ji, 2016*). Furthermore, ROS may influence the biology of the female reproduction system at the levels of the ovary, follicle, and oocyte (*Prasad et al., 2016*). The increased stress hormone concentrations such as cortisol reduce estradiol production by reducing the function of granulosa cell within the follicle, which lead to deteriorating oocyte quality (*Prasad et al., 2016*). Recently, we have reported that jumping exercise at high-intensity level could cause an increment in serum cortisol levels, and it was accompanied with lower levels of luteinizing hormone (LH), follicle stimulating hormone (FSH) and progesterone in adult female rats (*Mosavat, Ooi & Mohamed, 2014a*). It also has been found that an intense and exhaustive exercise program is accompanied by the reduction in uterus thickness in female rats (*Costa et al., 2014*).

Several non-pharmacological therapies such as increased caloric intake reduced exercise energy expenditure, and pharmacological treatment has been described to prevent and/or treatment of disorders caused by intense exercises (*De Souza et al., 2014*). Honey is a natural complex of sugars contains carbohydrates such as fructose, glucose, raffinose and sucrose, and flavonoids, enzymes, antioxidants, minerals, organic acids, proteins, phenolic acids, phytochemicals, and vitamins such as vitamins C and E (*Aljadi & Kamaruddin, 2004*). Tualang honey is a wild multi-floral honey found in the Malaysian Rain Forest with intermediate glycemic index as well as phenolic compounds and antioxidant activity (*Mohamed et al., 2010*), and the flavonoids present in honey particularly kaempferol and quercetin have been revealed to have estrogenic activity which may be beneficial for female reproductive health (*Jaganathan & Mandal, 2009*; *Oh & Chung, 2006*). It has been reported that administration of 0.2 g/kg Tualang honey elicited beneficial effects in the enhancement of uterus weight and thickness of uterus endometrium and vaginal epithelium in ovariectomized rats (*Zaid et al., 2010*). Furthermore, we have shown that administration of Tualang honey at one g/kg body weight could reduce the adverse effect induced by exercise on female reproductive hormones in rats (*Mosavat, Ooi & Mohamed, 2014a*, *2014b*). To date, however, whether different intensities of exercise may induce changes on histology of uterus and the possible protective effect of honey supplementation have not yet been reported.

Therefore, we aimed to investigate the effect of different jumping exercise intensities on uterus histology and the possible protective effect of honey supplementation in rats.

## MATERIALS AND METHODS

A total of 72, 9-week old female Sprague-Dawley rats with no significant mean difference in initial body weights (nearest 0.1 g) were entered in this study (*Mountjoy et al., 2014*; *Costa et al., 2014*). The experimental protocol was approved by the Animal Ethics Committee, Universiti Sains Malaysia (August 16, 2011, Reference: USM/Animal Ethics Approval/2011/(71) (325)), and has been described previously (*Mosavat, Ooi & Mohamed, 2014a*, *2014b*).

The estrous cycle is the main reproductive cycle of rodents that lasts 4 days and is categorized as: proestrus, estrus, metestrus (diestrus I) and diestrus (diestrus II), which can be determined according to the cell types observed in the vaginal smear (*Marcondes, Bianchi & Tanno, 2002*). The diestrus phase, which lasts 55–57 h and more than half of the cycle (*Hubscher, Brooks & Johnson, 2005*), is suitable for observing the population of different types of cells. To aim for standardization in the hormonal phase, the rats were scrutinized to detect diestrus phase two times; beginning and end of the experiment. The vaginal secretion of each rat was collected by flushing the vagina with one μL of normal saline (0.9%) using a clean pipette. The unstained wet smear was observed under a light microscope at 100× magnifications, similar pattern was used as standard for determining diestrus phase of the rats. The rats were assigned into six experimental groups (12 rats in each group) by block-randomization; control group with free cage activity and no intervention (C), low intensity; 20 jumps/day at 5 days/week for 8 weeks (20E), high intensity; 80 jumps/day for 5 days/week for 8 weeks (80E), honey supplementation for 7 days/week for 8 weeks (H), 20 jumps/day for 5 days/week combined with honey supplementation for 8 weeks (20EH), and 80 jumps/day for 5 days/week combined with honey supplementation for 8 weeks (80EH). Each rat in the training groups was positioned at the bottom of a specially designed wooden box and jumping was commenced by applying an electrical grid and after a few days of training, the rats jumped without electrical stimulation. Each jump took 4 s. The rats in the control group (C) were also handled during the duration of the study to imitate the stress induced by handling. At the end of the experiment, the rats were anesthetized using chloroform by lying for 2–3 min in a dried jar containing chloroform-soaked gauze pad and then they were decapitated in diestrus phase (Scientific Research Instrument, Oxford, UK) (*Mosavat, Ooi & Mohamed, 2014a*, *2014b*).

After decapitation, the uterus horns and ovaries were carefully removed and cleaned. The required organs were fixed in 10% neutral buffered formalin. These specimens were stored in a screw-capped specimen container prior to the next step procedures within a reasonable time. Each tissue was subjected to tissue fixation, processing, paraffin wax embedding, section cutting, and staining according to the previous study (*Suvarna, Layton & Bancroft, 2012*). Two histological slices were made from each organ for each rat. The standard hematoxylin and eosin (H & E) stained tissues of ovaries and uterine horns were observed for any changes under a light microscope attached with an image analysis system. Primary follicle consists of a central oocyte surrounded by a single layer of cuboidal granulose cells. The Graafian follicle is a large and mature follicle that has an antrum containing follicular fluid. Corpus hemorrhagicum is a temporary structure

formed immediately after ovulation from the ovarian follicle as it collapses and is filled with blood clot. Corpus luteum is formed after ovulation which consists of luteinized granulosa and theca cells. An ovarian luteal cyst is corpus luteum that has large cystic center. Endometrium is the tissue lining the inner cavity of the uterus and is bordered with simple epithelium line and comprises plentiful tubular glands. The myometrium is the middle layer of the uterine wall and comprised of a three smooth muscle layers, which are microscopically difficult to distinct. Uterine blood vessels is the myometrium that has strong and well-perfused vascular layer that is quite running around the uterus. The uterine gland or endometrial gland is a simple tubular gland shaped by invagination of the uterine endometrium. Qualitative and quantitative parameters of ovaries and uterine horns including the thickness of endometrium and myometrium, as well as the number of uterine glands and blood vessels were observed and measured. The mean of uterus thickness, including endometrium and myometrium of each rat was measured based on the maximum and minimum sagittal section thickness in μm. The number of uterine glands and blood vessels was counted once per slide for each rat and then the mean was calculated (Fig. 1). Regarding ovaries, the number of Graafian follicles, primary follicle, cysts, haemorregia, corpora lutea, and blood vessels were counted twice for each rat and then the mean was calculated (Fig. 2).

### Honey supplementation

The rats were fed with Malaysian Tualang honey at a dosage of one g/kg body weight/day by oral gavage for 7 days/week for 8 weeks. The rats in combined honey with jumping exercise groups (20EH and 80EH) were fed with honey, 30 min prior to the exercise session (*Mosavat, Ooi & Mohamed, 2014a*). The honey dosage was prescribed based on the rat biweekly body weight.

### Statistical analysis

The numerical data were studied using SPSS version 18.0. All variables were assessed for normality using the Normality Test as well as the Levene's Test to check for homogeneity of variance at 5% level of significance. One-way analysis of variance with post hoc test were performed to determine the significance of the differences between groups. The results are presented as mean ± standard error. The *p*-value of $<0.05$ was defined as statistically significant and used for all the comparisons.

## RESULTS

The data of body weights and reproductive organ weights presented in Table 1. The rats were weight matched at the beginning of the study, and there was no significant difference in weight gain percentage among all the experimental groups. There were also no significant differences in weight of ovarian among all the groups. Mean uterine weight in 20EH group was significantly greater compared to control, and there were no significant differences in uterine weight among all the experimental groups compared to control.

The quantitative histopathological findings and cross sections of uterine glands and blood vessels, the thickness of myometrium and endometrium are presented in Table 2;

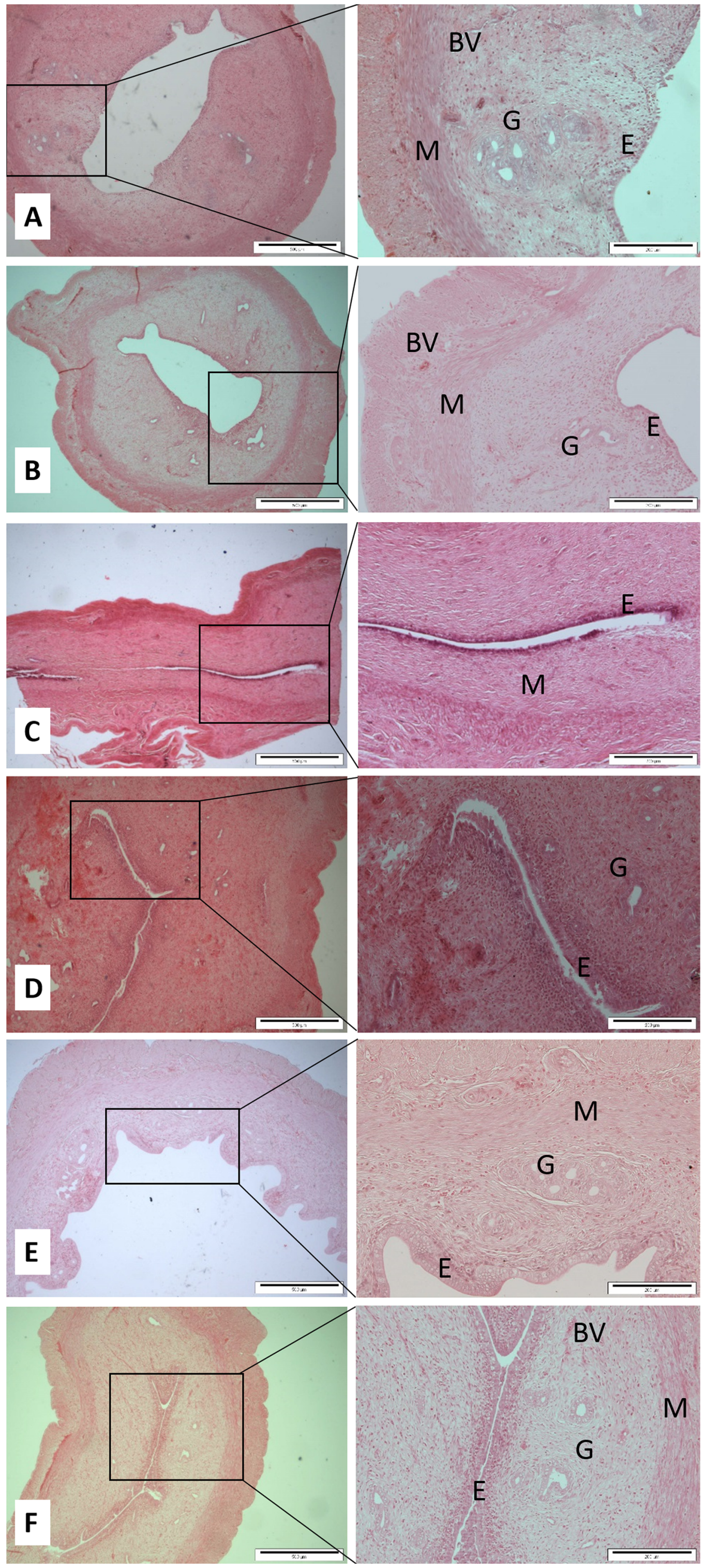

**Figure 1 Cross sections of the uterus (Magnification 4× with scale bar: 500 μm and 10× with scale bar: 200 μm) of (A) Control (C), (B) H (honey), (C) 20E (20 jumps/day), (D) 80E (80 jumps/day), (E) 20EH (20 jumps and honey supplementation), (F) 80EH (80 jumps and honey supplementation).** G, gland; M, myometrium; E, endometrium; BV, blood vessels.

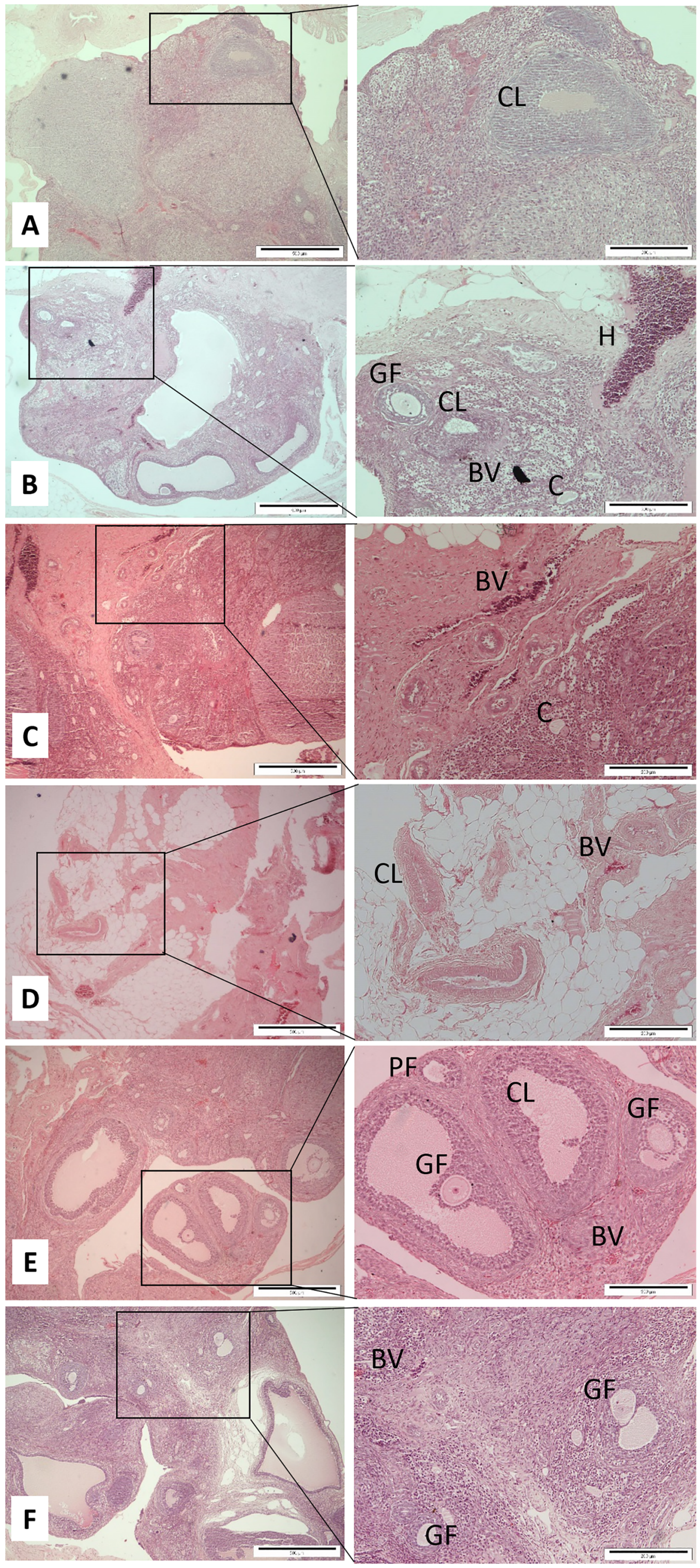

**Figure 2 Cross sections of the ovaries (Magnification 4× with scale bar: 500 µm and 10× with scale bar: 200 µm).** (A) Control (C), (B) H (honey), (C) 20E (20 jumps/day), (D) 80E (80 jumps/day), (E) 20EH (20 jumps and honey supplementation), (F) 80EH (80 jumps and. honey supplementation). PF, primary follicle; CL, corpus lutea; GF, graafian follicle; BV, blood vessels; C, ovarian luteal cyst; H, corpus hemorrhagicum.

Table 1 Body weights and reproductive organ weights (mean ± SE).

| Group | Initial body weight (g) | Weight gain % (g) | Ovarian weight (mg) | Uterine weight (mg) |
|---|---|---|---|---|
| C | 186.7 (16.5) | 76.2 (2.0) | 0.10 (0.02) | 0.41 (0.1) |
| H | 192.5 (16.3) | 76.6 (1.5) | 0.10 (0.03) | 0.46 (0.1) |
| 20E | 188.9 (10.5) | 75.4 (1.1) | 0.10 (0.02) | 0.48 (0.1) |
| 80E | 189.1 (9.3) | 75.8 (1.9) | 0.11 (0.02) | 0.48 (0.1) |
| 20EH | 191.6 (13.5) | 78.1 (2.3) | 0.10 (0.02) | 0.53 (0.3)[*] |
| 80EH | 193.3 (8.6) | 76.3 (1.8) | 0.10 (0.01) | 0.45 (0.1) |

**Note:**
[*] Significant from C ($p < 0.05$).

Table 2 Uterus quantitative histopathological findings and presented as mean ± SE.

| Groups | No. uterus glands | No. uterus vessels | Endometrium thickness (μm) | Myometrium thickness (μm) |
|---|---|---|---|---|
| C | 22.2 (1.9) | 8.4 (2.4) | 3.6 (0.5) | 107.1 (7.9) |
| 20E | 13.9 (1.2)[a] | 13.2 (3.9) | 3.6 (0.6) | 89.2 (6.6)[a] |
| 80E | 12.6 (2.1)[a] | 7.3 (1.7)[b] | 2.2 (0.4) | 79.9 (7.7)[a] |
| H | 21.9 (2.3)[b,c] | 18.5 (3.9)[c] | 7.5 (0.9)[a,b,c] | 96.3 (6.6) |
| 20EH | 21.6 (2.1)[b,c] | 19.3 (3.8)[c] | 6.3 (0.9)[a,b,c] | 121.9 (9.2)[b,c,d] |
| 80EH | 21.6 (3.6)[b,c] | 21.5 (2.8)[c] | 6.1 (0.5)[a,b,c] | 140.6 (8.2)[a,b,c,d] |

**Notes:**
[a] Significant from C ($p < 0.05$).
[b] Significant from 20E ($p < 0.05$).
[c] Significant from 80E ($p < 0.05$).
[d] Significant from H ($p < 0.05$).

Table 3 Ovary quantitative histopathological findings and presented as mean ± SE.

| Groups | Primary follicle | Graafian follicle | Corpora lutea | Ovarian luteal cysts | Corpus hemorrhagicum | Blood vessels |
|---|---|---|---|---|---|---|
| C | 4.1 (1.4) | 7.7 (1.4) | 8.4 (1.0) | 7.3 (1.3) | 2.2 (0.9) | 7.0 (1.3) |
| 20E | 2.2 (0.3) | 6.0 (0.6) | 6.7 (0.9) | 7.9 (2.1) | 0.3 (0.2)[a] | 7.2 (1.8) |
| 80E | 5.9 (1.5)[b] | 7.4 (1.4) | 8.8 (1.7) | 6.1 (2.0) | 1.8 (0.5)[a,b] | 6.0 (1.3) |
| H | 2.3 (0.4)[c] | 9.7 (1.0) | 8.1 (1.6) | 6.5 (1.5) | 0.2 (0.2)[a,c] | 11.9 (2.8)[c] |
| 20EH | 6.4 (1.1)[b,d] | 12.4 (1.6)[a,b,c] | 12.5 (2.3)[b,d] | 6.0 (1.9) | 0.0 | 15.1 (2.4)[a,b,c] |
| 80EH | 7.3 (1.6)[b,d] | 8.7 (1.7) | 7.0 (1.2)[e] | 7.6 (2.1) | 0.2 (0.2)[a,c] | 7.8 (2.5)[e] |

**Notes:**
[a] Significant from C ($p < 0.05$).
[b] Significant from 20E ($p < 0.05$).
[c] Significant from 80E ($p < 0.05$).
[d] Significant from H ($p < 0.05$).
[e] Significant from 20EH ($p < 0.05$).

Fig. 1, respectively. Mean number of primary and Graafian follicles, corpora lutea, cysts, haemorregia, blood vessels, and cross sections of the ovaries are presented in Table 3; Fig. 2, respectively.

The present data show that the 80 jumps/day were associated with lower endometrium thickness in comparison with the control group, however, results were not significant at 0.05 level. The endometrium was significantly thicker in the rats in H, 20EH, and 80EH groups compared to the rats in C, 20E, and 80E. The myometrium thicknesses of the rats in 80E were significantly lower compared to controls. However, the myometrium

thickness was significantly higher in 80EH compared to C. There was significantly higher myometrium thickness in 20EH and 80EH compared to 20E and 80E. Additionally, the myometrium thickness was significantly higher in 20EH and 80EH compared to H.

Regarding numbers of glands and blood vessels of the uterus, it was found that the numbers of glands of the uterus in 20E and 80E were significantly lower compared to C. However, there was a significantly high level of glands number in H, 20EH, and 80EH compared to 20E and 80E. The numbers of uterus blood vessels were significantly lower in 80E compared to 20E. However, the numbers of blood vessels were significantly higher in H, 20EH, and 80EH compared to 80E.

The histopathological finding of ovaries was revealed that the number of Graafian follicles did not differ between all experimental groups with exception of 20EH with significantly ($p < 0.05$) greater count of follicles compared to C, J20, and J80. The number of primary follicles ($p < 0.05$) were higher in 80E, 20EH, and 80EH compared to C. There were significantly ($p < 0.05$) higher of primary follicles in 20EH and 80EH compared to H. There was no significant difference in the number of ovarian cysts among the experimental groups. There was no count of haemorregia in 20EH group. The number of haemorregia were significantly ($p < 0.05$) lower in 20E, 80E, H, and 80EH compared to C. The number of haemorregia were significantly ($p < 0.05$) lower in 20E, H, and 80EH compared to J80. The number of corpora lutea were significantly ($p < 0.05$) lower in 80EH compared to H. However, the number of corpora lutea were significantly ($p < 0.05$) higher in 20EH compared to J20 and H. The number of ovary blood vessels was significantly ($p < 0.05$) higher in H compared to 80E. However, the levels of ovary blood vessels in HJ20 were significantly higher compared to C, J20, J80, and 80EH.

## DISCUSSION

This study firstly evaluated the probable histomorphometric changes of female reproductive organs influenced by different jumping exercise intensities. We observed that jumping exercise mainly 80 jumps/day (high-intensity exercise) induced negative effects on the measured histopathological parameters with significantly decreased in the number of uterus glands and myometrium thickness. Consequently, we observed that honey supplementation has played an effective role in diminishing these adverse effects induced by jumping exercise on uterus as well as improvement in ovary parameters characteristics.

It has been shown that intense exercise is commonly correlated with the boosted generation of ROS which may damage tissue and organ redox homeostasis (*Quindry, Kavazis & Powers, 2013*). Furthermore, it has also been shown that intense exercise-induced ischemic/reperfusion actions can be related to increased ROS production due to the deviation of the cardiac output to muscle mass inactivity and skin, consequently causing ischemia in the pelvis section (*Vollaard, Shearman & Cooper, 2005*). Our finding showed that honey supplementation may have potential in increasing the endometrium thickness in the rats feeding with honey alone or combined with 20 and 80 jumps/day compared to the rats in control and exercise alone groups. Similar effects of honey supplementation on reproductive organs have been reported in the study done by

*Zaid et al. (2010)* on ovariectomized rats, in which administration of honey supplementation for 2 weeks significantly increased the weight of the uterus and the thickness of vaginal epithelium. This previous study showed that improvement of uterus and vagina atrophy which might be attributed to the biologically active estrogen-like molecules or phytoestrogens exists in honey supplementation. In the present study, the myometrium thickness was significantly lower in 80 jumps/day group compared to the control group suggesting that there were negative effects of high exercise intensities on the female reproductive system. Previously, we have shown that high intensity of jumping exercises elicited negative effects on FSH, LH and progesterone concentrations in female rats which can possibly explain the changes in the thickness of endometrium and myometrium (*Mosavat, Ooi & Mohamed, 2014a*, *2014b*). It has been suggested by *Warren & Perlroth (2001)* that stress induced by exercise can arrest the gonadal function, through the increased glucocorticoids and catecholamines levels with activation of the corticotropin-releasing hormone neurons. Meanwhile, low caloric intake and high caloric expenditure, which can occur among athletes, suppress reproductive function (*Gibbs et al., 2011*), and this could be a suppressor factor for the gonadotropin-releasing hormone. In the present study, we observed that the myometrium was thicker in 20 and 80 jumps/day combined with honey supplementation groups compared to the control group and the rats with jumping alone, that is, 20 and 80 jumps/day groups. Additionally, the myometrium thickness was significantly higher in 20EH and 80EH compared to H. Regarding the number of glands and blood vessels of the uterus, it was found that the numbers of glands of the uterus in 20E and 80E were significantly lower than C, and the numbers of uterus blood vessels were significantly lower in 80E compared to 20E. These results implied that jumping exercise mainly high intensity with 80E/day may elicit negative effects on these measured parameters. We also found that there were increases in these measured parameters in jumping groups with honey supplementation compared to exercise groups without honey supplementation. Flavonoids contained in the honey with their antioxidant compounds may able to retard biologically damaging chemical reactions in living organisms through their ability to scavenge oxidants and free radicals (*Bertoncelj et al., 2007*). This positive effects of honey on endometrium may due to its high nutritional contents, particularly flavonoids which have antioxidant property (*Buratti, Benedetti & Cosio, 2007*) in scavenging ROS that may occur during exercise (*Hubscher, Brooks & Johnson, 2005*; *Suvarna, Layton & Bancroft, 2012*). Furthermore, based on *Oh & Chung (2006)* and *Jaganathan & Mandal (2009)* findings, flavonoids specially kaempferol, and quercetin are natural phytoestrogens as they show the estrogenic property. It also was reported that phytoestrogens were associated with an increased incidence of endometrial proliferation (*Unfer et al., 2004*). Furthermore, these positive effects of honey particularly in combination with jumping exercise on myometrium thickness, number of gland, and blood vessels may also be due to increase in supply of honey's components, that is, sugars, phenolic acid, and flavonoids through the increased of blood flow to the muscle (*Laughlin & Roseguini, 2008*) induced by jumping exercise and these vital elements contained in honey may have beneficial effects on uterus.

Our finding showed that high and low intensity jumping combined with honey supplementation were most effective in primary follicle generation in female rats.

This observation is similar to the findings of the study conducted by *Zaid, Othman & Kassim (2014)* in which the researcher showed that animals treated with Bisphenol A together with Tualang honey exhibited an improvement in the percentage of normal oestrous cycle compared to those treated with Bisphenol A. This finding may be attributed to the effect of honey on improving the normal oestrous cycle. The noteworthy histological findings of ovaries showed that low-intensity jumping exercise containing 20 jumps/day combined with one g/kg body weight of daily honey supplementation caused more obvious beneficial effects on the number of Graafian follicles, corpora lutea, and ovarian blood vessels formation compared to other groups in female rats.

The appearance of corpora lutea is considered the occurrence of ovulation. These results could be explained with this fact that exercise might promote follicular maturation and ovulation by decreasing sympathetic activity. The possible justification for the improvement of cyclicity is that reduced sympathetic activity caused by exercise may have a direct impact on the ovaries and sex steroid synthesis pathways (*Wu et al., 2014*). It has been shown that physical exercise decreased sympathetic nerve activity, improve menstrual frequency, and improved hyperandrogenism in women with the polycystic ovarian syndrome (*Benrick et al., 2013*).

The findings of this study showed that honey supplementation induced by different jumping exercise intensities caused no significant difference in the number of ovarian cysts in rats. The cystic follicle is formed from anovulatory follicle encircled by thin layers of granulosa cells with non-detectable theca cell layers (*Zaid, Othman & Kassim, 2014*). These findings suggest that exercise promotes follicular maturation and ovulation. The possible clarification for the improvement of cyclicity is that reduced sympathetic action may have a direct effect on the ovaries and female steroid production pathways. It has been reported that physical activity improved menstrual cycle through a reduction in sympathetic nerve activity and the levels of several sex steroids in women diagnosed with the polycystic ovarian syndrome (*Benrick et al., 2013*).

# CONCLUSIONS

In summary, this study demonstrated that honey has a protective effect against the histological and structural changes induced by jumping exercise on uterus and ovary in rats.

# ACKNOWLEDGEMENTS

We would like to thank the staff of Physiology and Sports Science Laboratories, School of Medical Sciences, Universiti Sains Malaysia.

## Funding

This work was supported by a short term grant provided by Universiti Sains Malaysia (No: 304/PPSP/61312031). The funders had no role in study design, data collection and analysis, decision to publish, or preparation of the manuscript.

## Grant Disclosures

The following grant information was disclosed by the authors:
Short Term Grant provided by Universiti Sains Malaysia: 304/PPSP/61312031.

## Competing Interests

The authors declare that they have no competing interests.

## Author Contributions

- Maryam Mosavat performed the experiments, analyzed the data, contributed reagents/materials/analysis tools, prepared figures and/or tables, authored or reviewed drafts of the paper, approved the final draft.
- Mahaneem Mohamed conceived and designed the experiments, performed the experiments, analyzed the data, contributed reagents/materials/analysis tools, prepared figures and/or tables, authored or reviewed drafts of the paper, approved the final draft.
- Foong Kiew Ooi conceived and designed the experiments, performed the experiments, analyzed the data, contributed reagents/materials/analysis tools, prepared figures and/or tables, authored or reviewed drafts of the paper, approved the final draft.
- Mitra Mirsanjari analyzed the data, approved the final draft, slides preparation.
- Anani Aila Mat Zin analyzed the data, prepared figures and/or tables, approved the final draft, reading the slides.
- Aminah Che Romli analyzed the data, approved the final draft, slides preparation.

## Animal Ethics

The following information was supplied relating to ethical approvals (i.e., approving body and any reference numbers):

The experimental protocol was approved by Animal Ethics Committee, Universiti Sains Malaysia (USM/Animal Ethics Approval 2011/ (71)(325)).

## Data Availability

The histological data is available in a Supplemental File.

## Supplemental Information

Supplemental information for this article can be found online at http://dx.doi.org/10.7717/peerj.7646#supplemental-information.

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
