# Peer review of "Histological changes of female reproductive organs subjected to different jumping exercise intensities and honey supplementation in rats"

_PeerJ, doi:10.7717/peerj.7646_

## Round 0.1 · original submission · Major Revisions

Your paper has been examined by 2 expert reviewers, each reviewer finds merit in the work but makes constructive suggestions. Please provide the animal ethical permission number with date.The introduction section needs to incorporate a paragraph with a brief description of the histological organization including the terms, which they will then use as primary follicle, cyst, haemorregia, blood vessels. The experimental design should be better defined, along with the statistical analyses applied; more data are needed to support your inferences. You need to improve the quality of your figures. Both reviewers give you suggestions to make your explanations about the honey supply more rigorous, and to adjust your discussion section. Please consider their suggestions carefully as they will help you to improve your manuscript considerably.

Reviewer 1 ·

Basic reporting

The article entitled 'Histological changes of female reproductive organs subjected to different jumping exercise intensities and honey supplementation in rats' possesses an interesting idea of research. But, it is not supported by valid experimental data. Several necessary information/data are missing. Authors are suggested to provide these data to support their hypothesis.

Experimental design

Authors should provide a few information/clarifications:

1. Please provide the animal ethical permission number with date.
2. Why authors have chosen 9-weeks old rats? Are these pubertal/adult rats? How would authors' like to justify the implementation of the findings in human females of same age group (pubertal/adult)? Please clarify.
3. Categorization based on estrous cycle phases are not clear. How authors have confirmed same estrous phase before initiating the study? Please mention.
4. Please provide the reference of using the honey supplement? Why authors have chosen 'Malaysian Tualang honey'?
5. What are also the basis of selecting 20 and 80 jumps respectively? Is there any reference supporting its usage as different intensities of exercise?

Validity of the findings

1. Only histopathological sections and morphometric data are not enough to support the findings. What are the causatives for these changes? Authors are suggested to provide the hormonal data of different groups of rats.
2. 80E seems to be very much deleterious to the ovarian structure. What about its hormonal milieu? Do the authors have the data for oxidative damage? If yes, please provide.
3. Some basic data, like body weight gain (%), reproductive organ weights are missing. Please provide those data.

Additional comments

The article entitled 'Histological changes of female reproductive organs subjected to different jumping exercise intensities and honey supplementation in rats' possesses an interesting concept, but not supported by enough experimental data. Authors are suggested to provide these data to support their hypothesis.

Reviewer 2 ·

Basic reporting

 The manuscript is clearly written. The English is unambiguous and the grammar is correct.
 The introduction contains enough information to understand the purpose of the manuscript. However, it would be important, for a better understanding, for those who are not fully familiar with the histology of the ovary and uterus, that the authors incorporate a paragraph where they briefly describe the histological organization including the terms, which they will then use as primary follicle, cyst, haemorregia, blood vessels, etc.
 Authors are encouraged to carefully read the instructions for authors on how to cite the literature reference in the manuscript.
 The manuscript presents the format of "standart sections" of journal.
 With regard to the Figures, it is first suggested that they be mentioned in the results. On the other hand, it is strongly recommended to improve the quality of the figures. Some of them are out of focus (Fig. 1A, AC, 1E, magnificaction 10x) or very dark (Fig. 2D 4X, 10x) or have a background colour that prevents the structures from being seen clearly (Fig. 1F, Fig. 2B). It would be advisable that in the photos with lower magnification (4x) the area where the highest magnification (10x) is going to be photographed be indicated with a box. Therefore, it is not necessary to include the magnification in each photo if there is a scale, which must be clearly added on the scale bar or in the legend of the figure.
 It is important that the authors clarify throughout the manuscript that when they use the term vessel they refer to blood vessels (lines: 120, 144, 157, 214. See document.doc).
 The authors should clarify in the manuscript, in the discussion section, the meaning of the abbreviation BPA (line 237).

Experimental design

 In relation to the background on the subject, the proposal of the authors is very interesting and original. It tries to answer questions raised by the group as a result of previous work on the proposed theme.
 The research protocol was approved by an institutional ethics committee.
 As for the section on materials and methods, it is not necessary for the authors to clarify the magnification used to carry out the observations (Line 118).
 It is important that they explain in the methodology in detail how they counted the uterine glands, blood vessels, follicles, etc., in the histological slices and avoid counting the same structure in the slices. In addition, in how many histological slices the measurements were made for each rat and how many measurements per rat were made for all the structures analyzed statistically (Line 121-126).

Validity of the findings

The conclusions expressed are appropriate according to the results obtained and the question posed. However, it is important that the authors review the statistical methodology.

Additional comments

I consider that the work is very valuable and novel, it only needs some corrections (statistics, improve the quality of photos, etc.) so that the work done by you are even better appreciated by the scientific community.

---

## Round 0.2 · accepted · Accept

Thank you for your consideration of our reviewers' suggestions. We are ready to move on.

Reviewer 2 ·

Basic reporting

Suggested corrections were made by the authors.

It is recommended that in the materials and methods section, the authors modify the way the fourth paragraph is written. Initially they mention the methodology used for processing the material for histological study and then mention the histological characteristics of the ovarian follicles and the uterus. I consider that they have to mention the characteristics of the different types of follicles and the histological organization of the uterus in another paragraph, explaining that these terms (types of follicles and layers of the uterus) are defined in order to have a histological criterion for later quantification.

The pictures of each figure have a better quality than the one initially presented.

Experimental design

The way in which some structures were measured was incorporated in the manuscript.
Modification has been made on the statement regarding ANOVA.

Validity of the findings

Modification has been made on the statement regarding ANOVA.

Additional comments

The modifications made in the manuscript will make this work appear by their scientific colleagues. Very good work!